# Expression, intracellular localization, and mutation of EGFR in conjunctival squamous cell carcinoma and the association with prognosis and treatment

Atsushi Sakai[1], Mizuki Tagami[1,2]*, Anna Kakehashi[3], Atsuko Katsuyama-Yoshikawa[2,4], Norihiko Misawa[1], Hideki Wanibuchi[3], Atsushi Azumi[2], Shigeru Honda[1]

1 Department of Ophthalmology and Visual Sciences, Graduate School of Medicine, Osaka City University, Osaka, Japan, 2 Ophthalmology Department and Eye Center, Kobe Kaisei Hospital, Kobe, Hyogo, Japan, 3 Department of Molecular Pathology, Graduate School of Medicine, Osaka City University, Osaka, Japan, 4 Division of Ophthalmology, Department of Surgery, Kobe University Graduate School of Medicine, Kobe, Hyogo, Japan

* tagami.mizuki@med.osaka-cu.ac.jp

**Data Availability Statement:** All relevant data are within the manuscript and Supporting Information files.

## Abstract

### Purpose

Conjunctival squamous cell carcinoma (SCC) is primarily treated with surgical resection. SCC has various stages, and local recurrence is common. The purpose of this study was to determine molecular localization of epidermal growth factor receptor (EGFR) and the possibility of EGFR as a biomarker for the management of conjunctival SCC.

### Methods

In this retrospective study, we performed immunohistochemistry to evaluate EGFR expression and localization in tumor cells, *EGFR* mutation-specific expression (E746-A750del and L858R), and human papillomavirus expression in a series of 29 conjunctival SCCs.

### Results

All 29 tumors in our cohort were EGFR positive (100%). Twenty-one of 29 tumors (72%) showed focal EGFR staining, and seven (28%) showed diffuse EGFR staining. In addition, we calculated the percentages of the two most important mutations in *EGFR* (exon 19 746-A750del (8/29, 27.5%), exon 21 (L858R mutant (2/29, 6.8%)) in conjunctival SCCs. We observed that the translocation of EGFR from the membrane into the cytoplasm was related to clinical prognosis, as we detected correlations between EGFR cytoplasmic staining and final orbital exenteration and between decreased EGFR membrane staining and progression-free survival.

**Funding:** None of the authors have any proprietary or financial interests to declare.

**Competing interests:** None of the authors have any proprietary or financial interests to declare.

## Conclusions

EGFR is important in the pathology of ocular surface squamous neoplasia including SCC and is a prognostic factor. Increased understanding of *EGFR* mutations may have important implications for future treatment options.

## Introduction

Ocular surface squamous neoplasia (OSSN) includes several diseases such as conjunctival pre-malignant dysplasia, carcinoma in situ, and invasive conjunctival squamous cell carcinoma (SCC) [1]. The annual incidence of OSSN was 0.53 cases/million/year (conjunctival intrae-pithelial neoplasia: 0.43 cases/million/year; SCC: 0.08 cases/million/year) in the United Kingdom [2, 3]. In the United States, the rate of SCC is 5-fold higher among males and whites [4].

Other previous research revealed that the risk increases with exposure to direct daylight and in outdoor workers. Meta-analysis demonstrated an association with human immunodeficiency virus (odds ratio, 6.2) and human papillomavirus (HPV) (odds ratio, 2.6) [4]. However, no large epidemiological studies have been performed on people living in the Far East.

Scholz et al. examined clinicopathological factors and biomarkers and identified promoter mutations in telomerase reverse transcriptase in 44% of 48 samples of conjunctival OSSN associated with ultraviolet light induction [5]. Recent research demonstrated that PD-L1 is expressed in almost half of conjunctival SCC cases and noted the potential application of immune checkpoint blockade as a treatment strategy for conjunctival SCC [6].

Molecular targeted therapy is now used to treat most carcinomas, and its use is continuing to increase [7]. Uveal melanoma also has recently been reported in the ocular oncology area [8]. Gefitinib is a relatively old tyrosine kinase inhibiter (TKI) that is used as a molecular targeted therapy, and its effects have been reported in various carcinomas. On the other hand, no basic clinical studies on ocular tumors have been reported [9–11]. In our current study, we investigated epidermal growth factor receptor (EGFR) expression in our cases to assess the possible effect of gefitinib. We also examined the molecular expression and intracellular localization of EGFR in conjunctival SCC in East Asian patients.

## Materials and methods

### Selection of cases and collation of clinicopathologic data

This study was approved by the Institutional Review Boards of Osaka City University and Kobe Kaisei Hospital and adhered to the tenets of the 1964 Declaration of Helsinki. Written informed consent was obtained from all patients before enrollment. We identified 29 patients treated by ophthalmologists (AA, MT) between November 2007 and July 2018 from whom we were able to procure tissue blocks with residual tumor. For each patient, we collected demographic features (age at initial diagnosis and at presentation to our institution, and sex) and primary tumor features (disease status at presentation [primary or recurrent] and in situ versus invasive disease). The American Joint Committee on Cancer (AJCC) stage, local recurrence (anatomic site and date), metastases (regional or distant and date), vital status at last follow-up, cause of death, types of surgery, and adjuvant therapy were also recorded.

### Immunohistochemistry (IHC)

Immunohistochemical studies for EGFR and HPV were performed on 6-μm-thick tissue sections using the following antibodies: anti-human EGFR rabbit monoclonal antibody (clone:

SP84; #414R-14; CELL MARQUE, Rocklin, CA, USA), anti-HPV mouse monoclonal antibody (clone: K1H8; ab75574; abcam, Cambridge, UK), horseradish peroxidase-conjugated anti-rabbit IgG (H+L) goat polyclonal antibody (HISTOFINE #424134, Nichirei Corporation, Tokyo, Japan), and horseradish peroxidase-conjugated anti-mouse IgG (H+L) goat polyclonal antibody (HISTOFINE #424144, Nichirei Corporation).

EGFR mutation-specific immunohistochemical staining was performed on 29 cases. As primary antibodies, we used EGFR E746-A750del (#2085, Cell Signaling Technologies, Danvers, MA, USA) and EGFR L858R (#3197, Cell Signaling Technologies), which were manually applied to the slides. Stained sections were viewed with an Olympus BX53+DP74.

As controls for staining, benign conjunctival lesions were also stained for EGFR, and colon cancer samples were stained as a positive control.

### Image analysis

Slides immunostained for EGFR, EGFR mutations, and HPV were evaluated in a blinded manner by two specialists (MT and AK). EGFR expression was visually estimated as the percentage of tumor cells with complete or partial membranous staining. Tumors with EGFR staining in ≥50% of tumor cells were considered the diffuse staining type (diffuse type), and those with <50% of tumor cells were considered the focal staining type (focal type). The presence or absence and intensity of cell membrane staining were semi-quantitatively divided into groups with a score of 0–3 (0: none, 1: weak, 2: strong, 3: very strong). The presence or absence and intensity of cell cytoplasmic staining were also divided into groups with a score of 0–3 and semi-quantitatively analyzed (0: none, 1: weak, 2: strong, 3: very strong). EGFR mutation-specific immunostaining was divided into two groups: those with immunostaining that was clearly present and those without immunostaining.

Slides immunostained for HPV were assessed with visual evaluation for the presence of punctate nuclear signals within tumor nuclei at 400× magnification and were scored as positive or negative.

### *EGFR* expression in tumors

*EGFR* expression in the tumor was analyzed with NanoString analysis. Archival formalin-fixed paraffin-embedded tumor tissue was retrieved and manually macrodissected. Total mRNA was isolated from the macrodissected tumor tissues using a Qiagen miRNeasy kit (Qiagen, Valencia, CA, USA) according to the manufacturer's instructions. The RNA sample was quantified with NanoDrop (Thermo Scientific, Wilmington, DE, USA) and regarded as adequate if it contained 400 ng at minimum. The sample was subsequently analyzed with the nCounter PanCancer Progression Panel (NanoString, Seattle, WA, USA) according to the manufacturer's instructions [12]. NanoString data processing was done with the R statistical programming environment (v3.4.2). Considering the counts obtained for positive control probe sets, raw NanoString counts for each gene were subjected to technical factorial normalization, which was carried out by subtracting the mean counts plus two times the standard deviation from the CodeSet inherent negative controls. Subsequently, biological normalization using the included mRNA reference genes was performed. Additionally, all counts with $P > 0.05$ after a one-sided t-test versus negative controls plus two times the standard deviation were interpreted as not expressed over basal noise.

### Statistical analysis

The clinical and histopathologic characteristics were summarized using descriptive statistics. Correlations between immunohistochemical, demographic, and clinicopathologic factors were

assessed using the Wilcoxon rank sum and Fisher's exact tests. Progression-free survival (PFS) was defined as the time from surgery to disease recurrence or death from any cause. Cox regression modeling was used to evaluate correlations between clinicopathologic and immunohistochemical features and survival outcomes. Statistical analyses were performed using SPSS Statistics version 22 software (IBM Japan, Tokyo, Japan). Values of $P < 0.05$ were considered statistically significant.

## Results

Clinicopathologic findings of our cohort are summarized in Table 1. All 29 patients in our cohort (100%) were East Asian, and included 15 men and 14 women with a mean age at presentation of 77.4 years. Fourteen patients (48%) had invasive SCC, and 15 (52%) had an in situ tumor. Primary orbital exenteration was necessary for local disease control in two patients (6%), and two patients (6%) underwent additional orbital exenteration. Nine patients (31%)

**Table 1. Clinicopathologic findings of 29 cases of conjunctival squamous cell carcinoma.**

|  | All (n = 29) n (%) |
|---|---|
| Age, years; mean (range) | 77.4 (63–98) |
| Sex |  |
| Male | 15 (52) |
| Female | 14 (48) |
| Follow-up after primary surgery; months (range) | 40.9 (3–135) |
| T-stage (AJCC) |  |
| Tis | 15 (52) |
| T1 | 3 (10) |
| T2 | 3 (10) |
| T3 | 7 (25) |
| T4 | 1 (3) |
| Primary surgery type |  |
| Local excision | 27 |
| Orbital exenteration | 2 |
| Adjuvant therapy |  |
| No | 20 (69) |
| Yes | 9 (31) |
| Additional excision | 7 |
| Topical chemotherapy | 1 |
| Radiation therapy | 1 |
| Immunohistochemical markers |  |
| HPV status in tumor cells |  |
| Negative | 22 (76) |
| Positive | 7 (24) |
| EGFR expression in tumors |  |
| Diffuse staining | 8 (27) |
| Focal staining | 21 (73) |
| Negative | 0 (0) |
| Cell membrane EGFR expression in tumors |  |
| Very strong | 1 (3) |
| Strong | 21 (72) |
| Weak | 7 (25) |

(*Continued*)

**Table 1.** (Continued)

| | All (n = 29) n (%) |
|---|---|
| Negative | 0 (0) |
| Cell cytoplasm EGFR expression in tumors | |
| Very strong | 4 (14) |
| Strong | 6 (20) |
| Weak | 19 (66) |
| Negative | 0 (0) |
| Outcome | |
| Orbital exenteration | |
| Yes | 4 (14) |
| No | 25 (86) |
| Local recurrence after curative therapy | |
| Yes | 9 (31) |
| No | 18 (69) |
| Metastasis | |
| Distant | 0 (0) |
| Regional + distant | 1 (3) |
| Regional | 1 (3) |
| None | 27 (94) |
| Vital status at last follow-up | |
| Dead | 2 (6) |
| Alive | 27 (94) |
| Cause of death | |
| Conjunctival SCC (metastasis) | 1 (50) |
| Other | 1 (50) |

underwent adjuvant therapy, most commonly additional local surgery. Topical chemotherapy and radiation therapy were performed in one patient in the adjuvant therapy group. Of this group, one patient died with disease 11 months after diagnosis of regional and lung metastases; the other patient was alive without disease at 44 months after diagnosis of regional metastases. Two patients (6%) died, one of which was due to conjunctival SCC (described above). Nine patients (31%) experienced local recurrence after curative surgery.

All 29 tumors were EGFR positive (100%) in our cohort. Twenty-one of 29 tumors (72%) showed focal EGFR staining, and seven (28%) showed diffuse EGFR staining (Fig 1). Analysis of EGFR intracellular staining patterns showed scores of 1.72 for membrane staining and 1.48 for cytoplasmic staining. No significant difference was found between carcinoma in situ (Tis) and invasive carcinoma (Tadv) (Table 2). No significant difference was found in the score depending on the stage. EGFR expression in colon cancer was used as a positive control (Fig 2A).

On the other hand, seven benign conjunctival lesions (three pinguecula, three pterygium, one dermoid cyst) showed partial weak positive staining in conjunctival squamous epithelial cells, especially on the cell membrane (Fig 2B). In addition, cytoplasmic staining was seen in only one case. Benign cases showed scores of 1.28 for membrane staining and 0.14 for cytoplasmic staining. Cytoplasmic staining patterns were significantly different in benign compared to SCC cases ($P < 0.01$) (Table 2). The correlation between EGFR staining (focal or diffuse) and EGFR localization (cytoplasmic staining group) was not significantly different,

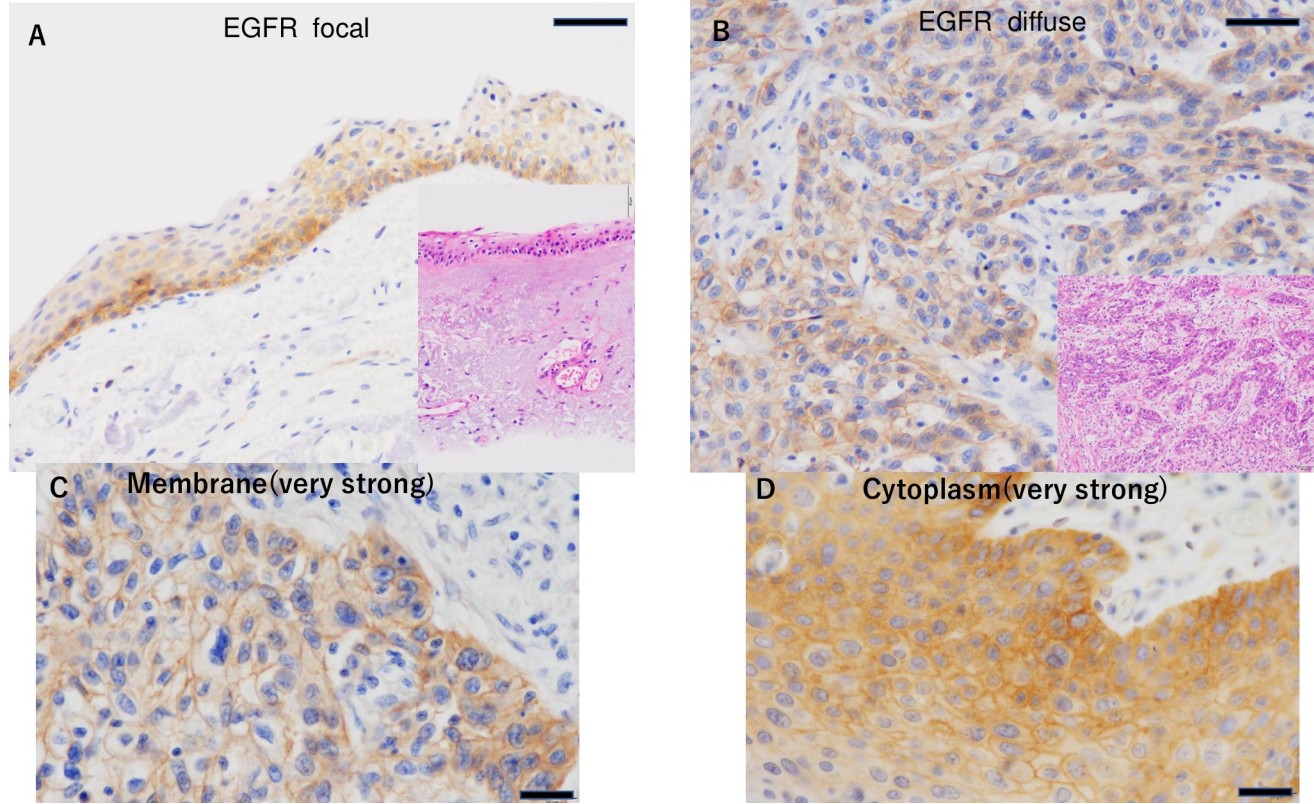

**Fig 1. EGFR expression in conjunctival SCC.** Focal EGFR staining (A) and diffuse EGFR staining (B) (Scale bar: 50 μm). Inset: corresponding field in a hematoxylin-eosin-stained section. Membrane staining (very strong: 3) (C) and cytoplasm staining (very strong: 3) (D) (Scale bar: 20 μm).

but the diffuse EGFR group tended to have a higher score ($p$ = 0.38 and 0.12, respectively) (Table 3).

EGFR E746-A750 del and EGFR L858R expression were assessed with immunohistochemistry in all 29 patients (Fig 3). The mutation at exon 19, *EGFR* E7446-A750 del, was confirmed in 8/29 (27.5%) cases, and that at exon 21, *EGFR* L858R point mutation, was confirmed in 2/29 (6.8%) cases with IHC (Table 4). The relationship between *EGFR* mutation and EGFR staining

**Table 2. Staining patterns of EGFR.**

|  |  | Staining patterns (N = 29) |  |  |  |  |  |
|---|---|---|---|---|---|---|---|
| Cell membrane |  | 0 | 1 | 2 | 3 | Average | $P$ |
| Tis (in situ) | N = 15 | 0 | 4 | 11 | 0 | 1.73 | 0.93 |
| Tadv (invasive) | N = 14 | 1 | 3 | 9 | 1 | 1.71 |  |
|  |  |  |  |  | total | 1.72 |  |
| Benign tumor | N = 7 | 1 | 4 | 2 | 0 | 1.28 | 0.30 |
| Cell cytoplasm |  | 0 | 1 | 2 | 3 | Average | $P$ |
| Tis (in situ) | N = 15 | 0 | 9 | 5 | 1 | 1.46 | 0.90 |
| Tadv (invasive) | N = 14 | 0 | 10 | 1 | 3 | 1.5 |  |
|  |  |  |  |  | total | 1.48 |  |
| Benign tumor | N = 7 | 6 | 1 | 0 | 0 | 0.14 | <0.01* |

*p value based on the non-paired t-test.

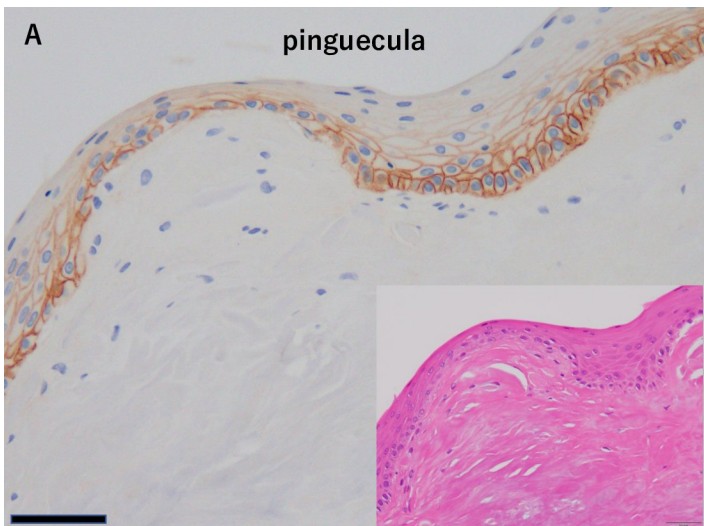
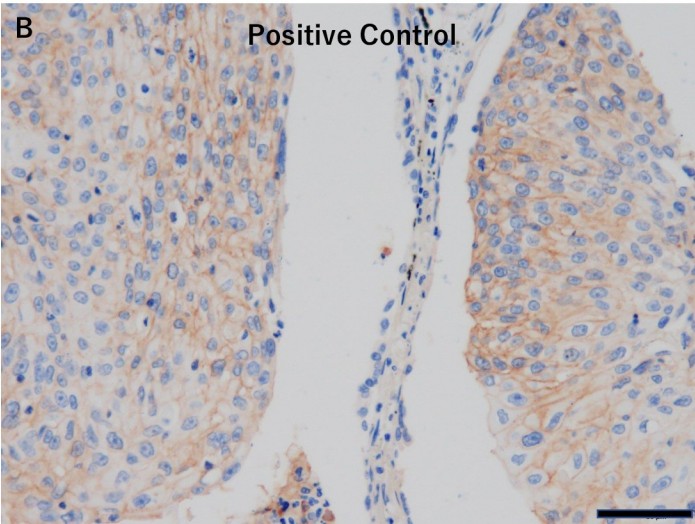

**Fig 2.** (A) EGFR expression in colon cancer as a positive control (Scale bar: 50 μm). (B) EGFR expression in a control benign lesion, pinguecula (Scale bar: 50 μm). Inset: corresponding field in a hematoxylin-eosin-stained section.

(focal or diffuse) was determined using univariate linear regression analysis with correction for age ($P = 0.559$).

Regarding *EGFR* expression in tumors, we compared the Tis and Tadv groups according to AJCC T grading (n-4, 4). No significant difference was found ($P = 0.162$) (Fig 4).

The majority of patients in our cohort were HPV negative (n = 22; 75%) (Table 1). The positive rate of HPV immunoreactivity increased with increases in AJCC T grading, but the correlation was not statistically significant.

The Cox regression model was used to examine and analyze the relationship between long-term prognosis including orbital exenteration and PFS and the clinicopathological status, EGFR staining pattern, and *EGFR* mutation. Univariate Cox regression analyses revealed significant correlations between EGFR cytoplasmic staining and final orbital exenteration (hazard ratio (HR): 4.2, $P = 0.036$) (Table 5). Additionally, a significant correlation was seen between the T stage (AJCC) and PFS, and between EGFR membrane staining and PFS (HR: 13.1, 0.23, $P = 0.025$, $P = 0.015$, respectively) (Table 6). Local recurrence, distant metastasis rate, and overall survival rate were not statistically significant. In addition, the *EGFR* mutation was not significantly correlated with final orbital exenteration or PFS (Tables 5 and 6).

## Discussion

To the best of our knowledge, this is one of the first studies to survey the prevalence of *EGFR* mutations and intracellular localization in conjunctival SCC and to evaluate the prognostic significance of tumor cells that express EGFR in conjunctival SCC.

In this study, we found that the tumor tissue of all conjunctival SCCs (100%) expressed EGFR. In addition, we determined the percentages of the two most important mutations in

**Table 3. EGFR staining and localization.**

|  | EGFR Focal(N = 21) | EGFR Diffuse(N = 8) | *P* |
|---|---|---|---|
| Cell Membrane | 1.8±0.9 | 1.5±0.9 | 0.38 |
| Cell cytoplasmic | 1.3±0.6 | 1.8±0.8 | 0.12 |

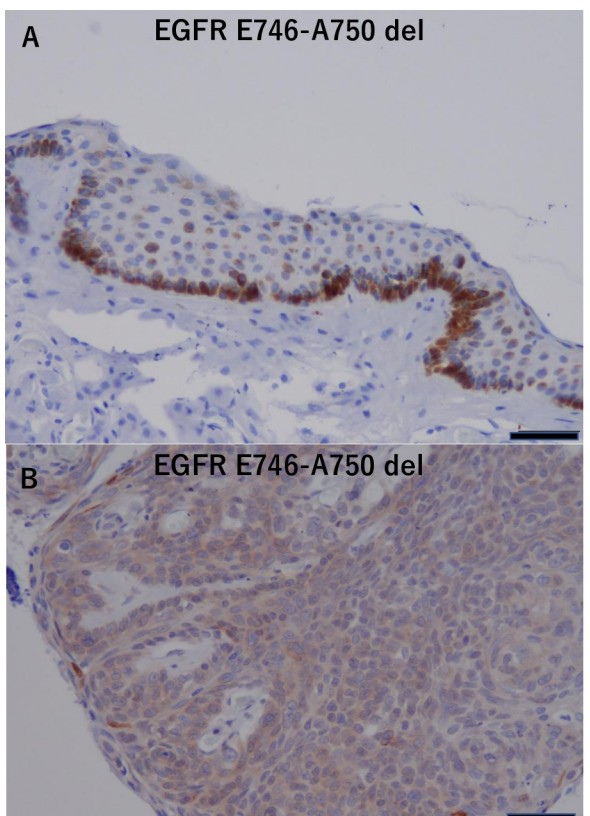

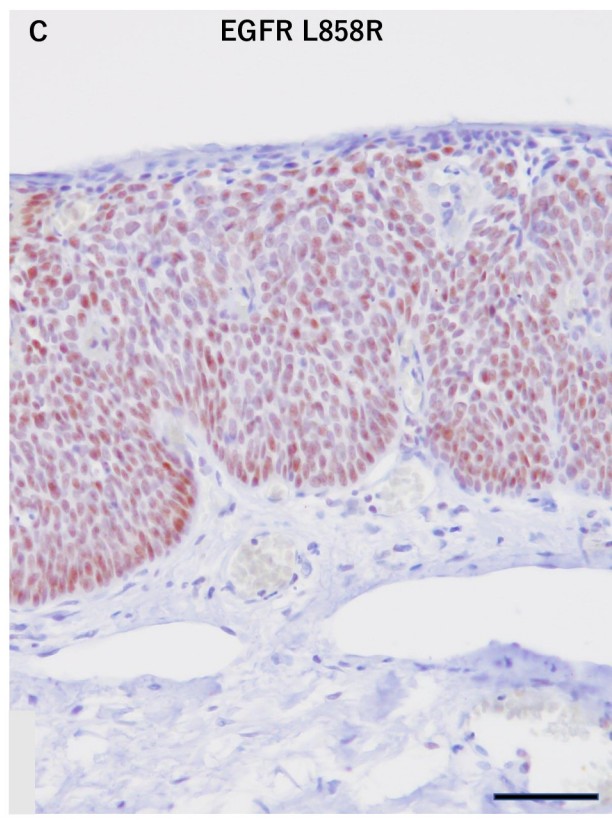

**Fig 3. *EGFR* mutation-specific expression in conjunctival SCC.** (A) Basement membrane staining in a tumor with *EGFR* E746-A750 del. (B) Whole tumor staining in an *EGFR* E746-A750 del mutant. (C) Conjunctival SCC layer cells with strong staining in an *EGFR* L858R mutant (Scale bar: 50 μm).

EGFR (exon 19 746-A750del (8/29, 27.5%), exon 21 (L858R Mutant (2/29, 6.8%)) in conjunctival SCCs. We also showed that the translocation of EGFR from the membrane into the cytoplasm was related to clinical activation of cancer, as correlations between EGFR cytoplasmic staining and final orbital exenteration, and between decreased EGFR membrane staining and PFS were noted. Although the number of cases examined was small, the expression of cytoplasmic staining of EGFR was weak but significantly different from membrane staining in the benign disease group. Our hypothesis is that as EGFR transitions from the membrane into the cytoplasm, malignant changes progress. In addition, a correlation between EGFR staining (focal or diffuse) and EGFR cytoplasmic staining was seen, and a higher score tended to be present in the diffuse EGFR staining group.

**Table 4. Summary of *EGFR* E746-A750 del and *EGFR* L858R point mutations.**

| Mutation | N = 29 (%) | Age (y) | Sex (M/F) | T stage | EGFR staining patterns (diffuse/focal) | EGFR localization score (membrane/cytoplasmic) |
|---|---|---|---|---|---|---|
| Exon 19 *EGFR* E746-A750 del (N = 8) | 8/29 (27.5%) | 75.8 | 3/5 | T3: 4 | 2/6 | 1.6/1.5 |
| | | | | T2: 2 | | |
| | | | | Tis: 2 | | |
| Exon 21 *EGFR* L858R point mutation (N = 3) | 2/29 (6.8%) | 73.0 | 1/1 | T3: 1 | 1/2 | 1/3 |
| | | | | Tis: 1 | | |

M: Male, F: Female.

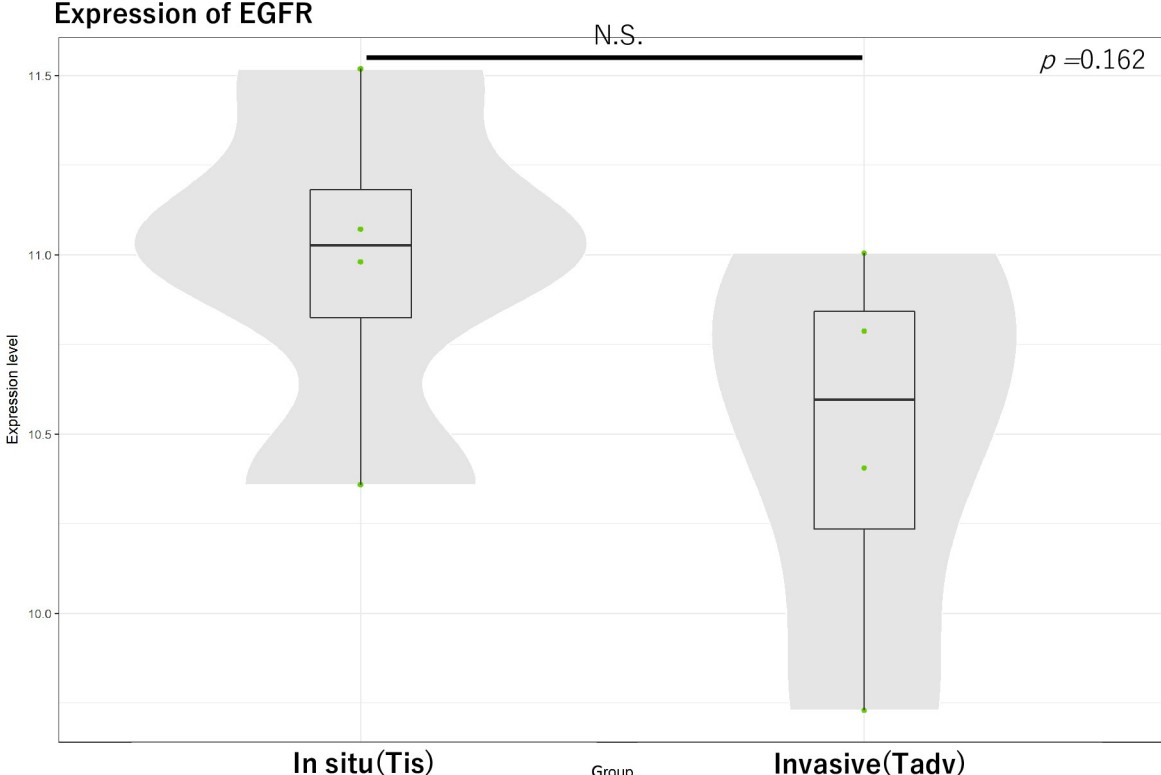

**Fig 4. For *EGFR* expression in tumors, we compared carcinoma in situ (Tis) and invasive carcinoma (Tadv) groups according to AJCC T grading (n-4, 4).** N.S., not significant.

Intracellular transfer of EGFR in the group with diffuse staining may indicate progression, and although no statistical differences were observed in this study, significant findings may emerge by increasing the number of cases in the future.

In the past, especially in African countries, several studies on conjunctival SCCs and EGFR expression have been reported. They suggested a potential association with HPV [13, 14]. Other previous studies reported that post-translational modification can promote EGFR

**Table 5. Relationship between orbital exenteration and clinicopathologic and molecular factors.**

| Univariate analysis | | | | |
|---|---|---|---|---|
| Variables | N = 29 | HR | 95% CI | P |
| Age | Mean 77.3 years | 1.68 | 0.834–3.406 | 0.146 |
| Sex | Male 15, Female 14 | 0.925 | 0.129–6.605 | 0.938 |
| T-stage (AJCC) | Tis, T1, T2: 21/T3 ≥8 | 7.551 | 0.785–72.650 | 0.080 |
| **EGFR** staining | Focal 21/Diffuse 8 | $7.21 \times 10^2$ | $0.001–82.9 \times 10$ | 0.365 |
| **EGFR** membrane staining | Very strong 1/Strong 21/Weak 7/Negative 0 | 0.415 | 0.121–1.429 | 0.164 |
| **EGFR** cytoplasmic staining | Very strong 4/Strong 6/Weak 19/Negative 0 | <u>4.206</u> | 10.97–16.122 | 0.036* |
| *EGFR* mutation | Exon 19 E746-A750 del 8/Exon 21 L858R point mutation 2 | 0.582 | 0.096–3.545 | 0.558 |
| HPV positive | Positive 7/Negative 22 | 0.032 | $0.00–5.07 \times 10^2$ | 0.485 |

CI indicates confidence interval; HR hazard rate.

Statistically significant differences are underlined.

*p value based on the Cox proportional hazard model.

**Table 6. Relationship between PFS and clinicopathologic and molecular factors.**

| Univariate analysis | | | | |
|---|---|---|---|---|
| Variables | N = 29 | HR | 95% CI | *P* |
| Age | Mean, 77.3 years | 1.15 | 0.970–1.384 | 0.104 |
| Sex | Male 15/Female 14 | 0.611 | 0.101–3.690 | 0.592 |
| T-stage (AJCC) | Tis, T1, T2 21/T3 ≥8 | <u>13.11</u> | $1.384–1.24 \times 10^2$ | 0.025* |
| **EGFR** staining | Focal 21/Diffuse 8 | 3.635 | 0.685–19.289 | 0.130 |
| **EGFR** membrane staining | Very strong 1/Strong 21/Weak 7/Negative 0 | <u>0.237</u> | 0.074–0.759 | 0.015* |
| **EGFR** cytoplasmic staining | Very strong 4/Strong 6/Weak 19/Negative 0 | 2.813 | 0.993–7.973 | 0.052 |
| *EGFR* mutation | Exon 19 E746-A750del 8/Exon 21 L858R point mutation 2 | 33.512 | $0.00–1.91 \times 10^7$ | 0.604 |
| HPV positive | Positive 7/Negative 22 | 0.459 | 0.052–4.077 | 0.484 |

CI indicates confidence interval; HR hazard rate.

Statistically significant differences are underlined.

*p value based on the Cox proportional hazard model.

endocytosis and lysosomal degradation of EGFR, thereby ensuring termination of receptor signaling [15, 16].

In our cohort, expression and localization of EGFR and its association with prognosis were first reported in the Asian race. Additionally, intracellular translocation of EGFR from membrane staining to cytoplasm staining, likely by endocytosis, was associated with the percent of final orbital exenteration (cytoplasmic staining HR: 4.206, $P < 0.036$) and PFS (membrane staining HR: 0.237, $P < 0.015$) in our cohort. Regarding the difference in local changes in EGFR immunoreactivity in patients without *EGFR* expression in the tumor, we compared the Tis and Tadv groups according to AJCC T grading. A recent study showed that feedback regulatory loops can modulate growth factors and receptor tyrosine kinases such as EGFR to regulate cellular functions including abnormal states such as cancer [17]. Our study examined this phenomenon clinically and confirmed a pathological difference without changes in gene expression.

*EGFR* mutations in OSSN including invasive SCCs have not been examined in Asian patients. Since 2016, approximately 16,000 *EGFR* mutations in lung cancer had been registered in the COSMIC (the catalog of somatic mutations in cancer) database. Most (93%) are concentrated in the exon 18–21 region of the intracellular tyrosine kinase domain. The most frequent one is at codon 746 of exon 19. A deletion mutation is present at a site centered on five amino acids (ELREA) near amino acid 750, and a point mutation changes leucine to arginine (L858R) at codon 858 of exon 21 [18]. Shigematsu et al. in 2005 and Mitsudomi et al. in 2007 reported that *EGFR* mutations are common in Asians, females, non-smokers, and adenocarcinomas in lung cancer [19, 20]. Generally, when *EGFR* mutation occurs, the tyrosine kinase activity of EGFR at the ATP binding site is constantly active, even without growth factor. Cancer cell growth and survival depend on this pathway (oncogene addiction). EGFR TKIs competitively inhibit ATP binding in the kinase domain and suppress autophosphorylation of EGFR. Blockade of signal transmission has antitumor effects [21]. Previous reports of *EGFR* activating mutations (common mutations) described the frequency of exon 19 deletion mutations as 44.8% (2573/5741) and 39.8% for L858R mutations (2283/5731) in lung cancer [18, 22, 23].

*EGFR* mutations were examined to verify the effect of gefitinib on positive non-small cell lung carcinoma in two Phase III clinical trials from Japan. In the NEJ002 trial and the WJTOG3405 trial, gefitinib was the test treatment group; the standard treatment in the former

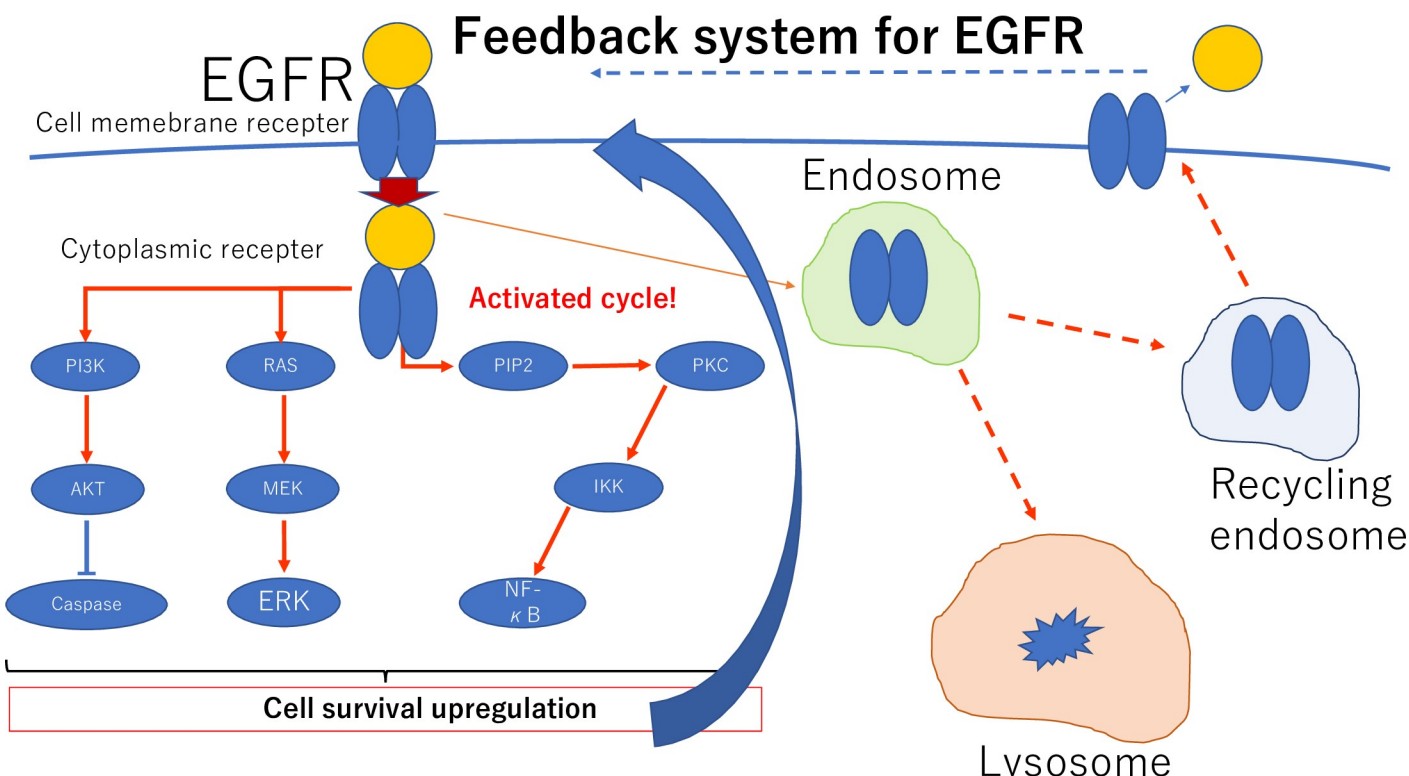

**Fig 5. Schematic of movement of EGFR into the cytoplasm by endocytosis to avoid excessive signaling and for recycling.**

was carboplatin + paclitaxel, and in the latter was cisplatin. In all studies, the gefitinib group showed superior PFS [24, 25]. In view of these findings in lung cancer in Asians, our findings regarding EGFR expression and mutations will provide further options for potential treatment of OSSN for pre- and post-surgical treatment.

The association of SCC with HPV was not confirmed because the number of cases was small. In addition, our results may not be accurate because we did not use multiplex PCR, which is currently the most suitable genotyping method [26].

Ours is the first report to show that differences in the expression form and mutations in EGFR in OSSN are associated with prognosis and treatment.

In an animal model, EGFR inhibition affected epithelial cell proliferation and stratification during corneal epithelial wound healing and may play a role in maintaining normal corneal epithelial thickness [27].

Gefitinib is an EGFR inhibitor and is the first approved molecular targeted therapy for cancer treatment in Japan [28]. Thus, understanding the pathological role of EGFR in OSSN and applying it to treatment are of great significance for seeking new treatment indications in OSSN including conjunctival SCCs. In this study, EGFR may translocate from the cell membrane into the cytoplasm. Tumor cells may transfer EGFR into the cytoplasm by endocytosis to avoid excessive signaling by the feedback system (Fig 5) [29]. Furthermore, in this study, the EGFR mutation was present in many patients with OSSN. This finding may suggest a course of treatment in the future. In addition, the method we used for identification of *EGFR* mutations was not general genotyping, but was a judgment of immunohistochemically stained sections. Although the sensitivity and specificity were high in a previous report, this is still a limitation [30].

This study has important limitations. First, regarding EGFR expression on the ocular surface, changes in benign diseases and age-related changes in normal tissues may not have been sufficiently investigated. Our study found that *EGFR* mutations were also present in conjunctival SCC in east Asians. However, we did not obtain results that correlated with the final prognosis. Further studies including further multi-institutional studies and an increase in the number of cases will be needed in the future. Another limitation is that double testing of formalin-fixed paraffin-embedded specimens and plasma with real-time PCR for detection of *EGFR* mutations is more common than IHC in actual clinical practice. According to the literature, both the sensitivity and specificity were satisfactory for these two types of mutations [30]. In addition, the size of our study cohort was small (n = 29), and the length of follow-up (less than 1 year in some patients) may not have been sufficient for long-term outcome analyses. Therefore, additional studies will be needed to corroborate our findings.

In conclusion, the results of this study indicate that EGFR is an active molecular target in the pathology of OSSN including SCC and is a prognostic factor. The finding also suggests that discovery of mutations may have important implications for future treatment options.

## Supporting information

**S1 File.**
(XLSX)

## Acknowledgments

We gratefully acknowledge the technical assistance of the Research Support Platform, Osaka City University Graduate School of Medicine and the Clinical Laboratory Department of Kobe Kaisei Hospital.

## Author Contributions

**Conceptualization:** Mizuki Tagami, Atsushi Azumi.

**Data curation:** Atsushi Sakai, Mizuki Tagami, Atsuko Katsuyama-Yoshikawa, Norihiko Misawa, Atsushi Azumi.

**Formal analysis:** Mizuki Tagami, Anna Kakehashi, Norihiko Misawa.

**Funding acquisition:** Mizuki Tagami.

**Investigation:** Mizuki Tagami, Atsuko Katsuyama-Yoshikawa, Atsushi Azumi.

**Methodology:** Mizuki Tagami, Anna Kakehashi, Atsuko Katsuyama-Yoshikawa, Atsushi Azumi.

**Project administration:** Mizuki Tagami.

**Supervision:** Anna Kakehashi, Hideki Wanibuchi, Atsushi Azumi, Shigeru Honda.

**Visualization:** Atsushi Sakai, Mizuki Tagami, Anna Kakehashi.

**Writing – original draft:** Atsushi Sakai, Mizuki Tagami.

**Writing – review & editing:** Mizuki Tagami, Shigeru Honda.

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
