## [Decision Letter · Decision Letter 0]

20 May 2020

PONE-D-20-09283

Expression, intracellular localization, and mutation of EGFR in conjunctival squamous cell carcinoma associated with prognosis and treatment

PLOS ONE

Dear Dr. Tagami,

Thank you for submitting your manuscript to PLOS ONE. After careful consideration, we feel that it has merit but does not fully meet PLOS ONE’s publication criteria as it currently stands. Therefore, we invite you to submit a revised version of the manuscript that addresses the points raised during the review process.

All learned reviewers have provided critical comments that need to be appropriately addressed.

We would appreciate receiving your revised manuscript by Jul 04 2020 11:59PM. To enhance the reproducibility of your results, we recommend that if applicable you deposit your laboratory protocols in protocols.io, where a protocol can be assigned its own identifier (DOI) such that it can be cited independently in the future. For instructions see: http://journals.plos.org/plosone/s/submission-guidelines#loc-laboratory-protocols

We look forward to receiving your revised manuscript.

Kind regards,

Sanjoy Bhattacharya

Academic Editor

PLOS ONE

Journal Requirements:

2. Please note that PLOS does not permit references to “data not shown.” Authors should provide the relevant data within the manuscript, the Supporting Information files, or in a public repository. If the data are not a core part of the research study being presented, we ask that authors remove any references to these data.

3. For studies involving humans categorized by race/ethnicity, age, disease/disabilities, religion, sex/gender, sexual orientation, or other socially constructed groupings, authors should update outmoded terms and potentially stigmatizing labels to more current, acceptable terminology. Example: “Whites” should be changed to “people of [Western] European descent” (as appropriate).

4. In the ethics statement in the manuscript and in the online submission form, please provide additional information about the tissue samples used in your retrospective study. Specifically, please provide the source of the tissue samples analyzed in this work (e.g. hospital, institution or medical center name). If patients provided informed written consent to have data and tissue samples from their medical records used for research purposes, please include this information.

5. Please include your IRB/ethics committee approval numbers in your ethics statement

Additional Editor Comments (if provided):

Reviewers' comments:

Reviewer's Responses to Questions

**Comments to the Author**

1. Is the manuscript technically sound, and do the data support the conclusions?

Reviewer #1: Partly

Reviewer #2: Partly

Reviewer #3: Partly

2. Has the statistical analysis been performed appropriately and rigorously? 

Reviewer #1: Yes

Reviewer #2: Yes

Reviewer #3: I Don't Know

3. Have the authors made all data underlying the findings in their manuscript fully available?

Reviewer #1: Yes

Reviewer #2: Yes

Reviewer #3: Yes

4. Is the manuscript presented in an intelligible fashion and written in standard English?

Reviewer #1: Yes

Reviewer #2: Yes

Reviewer #3: Yes

5. Review Comments to the Author

Reviewer #1: This is an interesting study that evaluates EGFR status (intracellular localization and mutation spectrum) in conjunctival squamous carcinoma. Trying to fill a void in the basic science research on conjunctival SCC, the paper fails to adequately address the aims of the study.

Comments:

1. HPV status, role in tumorigenesis. The authors are right in their conclusion that the number of cases was small but the manner in which they assessed HPV infection needs revision. They reference to several studies on conjunctival SCC and HPV but the 2 papers they cite are on HIV positive patients of African descent. In the current era of understanding the role of high and low risk HPV in human tumorigenesis and differential testing for various HPV genotypes, the manner they chose to assess HPV, by IHC with a (probably) very broad HPV antibody lacks the desired specificity. The authors should consider either eliminating this part of the study or reevaluation of HPV status based on more recent literature pertinent to the immunocompetent subjects and in conjunction with employing more specific methods.

2. The authors use immunohistochemistry staining to draw conclusions on the intracellular distribution of EGFR and the translocation of EGFR from the membrane into the cytoplasm. In table 2 they show that all cases of invasive SCC and 14/15 cases of in situ carcinoma show various degrees (weak in the majority of cases) of cytoplasmic staining in combination with membranous staining (14/14 in invasive SCC, 14/15 in situ carcinoma). There is no correlation between the degree/intensity of membranous vs cytoplasmic staining. Assuming that cytoplasmic staining marks translocation of EGFR from the membrane into cytoplasm is not scientifically sound.

3. The authors claim that this is the first report to show that mutations in EGFR are associated with prognosis and treatment. They allude to the EGFR mutations in lung cancer and they show in table 3 the results of the 2 investigated mutations but there is no correlation of the 2 mutations with prognosis and treatment in ocular neoplasia.

4. The specificity of the 2 antibodies used for the 2 EGFR mutations is not addressed and no correlation with molecular studies identifying the same 2 mutations is done. However, the authors conclude that IHC can be used to identify EGFR mutations.

Reviewer #2: In the manuscript entitled " Expression, intracellular localization, and mutation of EGFR in conjunctival squamous cell carcinoma associated with prognosis and treatment” Sakai A. et al attempted to identify molecular localization of epidermal growth factor receptor (EGFR) and its role as novel biomarker for the management of conjunctival SCC. Authors used immunohistochemistry approach to evaluate EGFR expression and localization. They also used similar approach to visualize mutation associated with EGFR in SCC. Authors further implicated the translocation of EGFR from the membrane into the cytoplasm is prognostic marker for conjunctival squamous cell carcinoma. This is an interesting clinical study which focusses only east Asian population, however, a few major issues need to be addressed before the manuscript gets ready for the publication. The major concerns are as follows:

1. Authors have first found out that EGFR have focal and diffuse staining in patients. This is really an interesting observation however, authors later never looked for its correlation with localization or mutation of EGFR in patients.

2. In figure 1A&B, authors showed representative images of local and diffuse EGFR staining. It appears that images of EGFR staining in both figures have different magnification. Similarly figure 1E captured at different magnification compared to figure 1C&D. Figure 2 have similar issue where image D have different magnification compared to rest of the images. Authors need to make sure representative images were captured at same magnification for better comparison. Authors also need add scale bar on all the microscopy images.

3. In figure 1A hematoxylin-eosin-stained section in inset looks little different from EGFR stained image. Authors also need to make sure that the image in H&E inset properly corresponds to EGFR stained image.

4. In present manuscript authors also evaluated EGFR mutation using immunohistochemistry. They observed EGFR E746-A750 del and EGFR L858R expression in patients. Authors need to include this parameter for univariate correlation analysis with PFS and orbital exenteration. Also, it will be interesting to see the effect of these mutation on the localization of EGFR in tumor cell.

5. Authors also need to perform multivariate analysis for clinical correlation for both membrane and cytoplasmic EGFR localization in patients.

6. References in the manuscript have different fonts and wrongly numbered. Authors need to use endnote or reference manager to add the references.

Reviewer #3: In the submitted manuscript, Sakai et al evaluated the expression, localization, and mutation of EGFR in conjunctival squamous cell carcinoma. Using immunohistochemistry, they observed that all 29 samples were EGFR positive and classified its expression into membrane, cytoplasm, focal, and diffuse. Eight samples presented the E746-A750 del mutation while 2 samples the L858R point mutation. They also found a correlation between EGFR cytoplasmic staining and orbital exenteration, and between decreased EGFR membrane staining and progression-free survival. Although, the role of EGFR has been extensively study in many different cancer types, little is known about EGFR in conjunctival squamous cell carcinoma. The current study brings valuable information to the field and is on a topic of relevance. However, some concerns must be addressed before further consideration.

Specific comments

1. The result section should be written in more detail for easier reading.

2. Line 154: “Analysis of EGFR intracellular staining patterns showed scores of 1.72 for membrane staining and 1.48 for cytoplasmic staining (Figure 1)”. This result is found in Table 2 not in figure 1.

3. Table 2: Revise statistics. My calculation doesn’t match the average shown in the table.

4. Is there any difference in EGFR expression localization (membrane or cytoplasm) between focal and diffuse staining?

5. Line 156: “Seven benign conjunctival lesions were partially weak positive in conjunctival squamous epithelial cells near basement membrane”. Specify the total number of benign samples evaluated (7 out of 7?). It would be interesting to show the score analysis of benign samples to compare to tumor samples.

6. Line 157: “We found no significant changes between Tis (in situ) and Tadv (SCC advanced cases) (Table 2)”. Specify what changes.

7. Line 159: Include more information regarding EGFR E746-A750 del and EGFR L858R expression like diffuse/focal staining, membrane/cytoplasmic score. Is there any correlation between EGFR mutation and clinical data (orbital exenteration/PFS)?

8. Table 1: EGFR expression in tumors - diffuse staining (8) and focal staining (19). Are these numbers correct, total samples 27?

9. Figure 1 and 2: When comparing images use the same magnification. Verify if scale bars are correct.

6. PLOS authors have the option to publish the peer review history of their article (what does this mean?). If published, this will include your full peer review and any attached files.

Reviewer #1: No

Reviewer #2: No

Reviewer #3: No

---

## [Author Response · Author response to Decision Letter 0]

12 Jun 2020

Response to reviewers 

Reviewer #1: Thank you for your valuable comments.

Our responses to your comments are below.

Comments:

1. HPV status, role in tumorigenesis. The authors are right in their conclusion that the number of cases was small but the manner in which they assessed HPV infection needs revision. They reference to several studies on conjunctival SCC and HPV but the 2 papers they cite are on HIV positive patients of African descent. In the current era of understanding the role of high and low risk HPV in human tumorigenesis and differential testing for various HPV genotypes, the manner they chose to assess HPV, by IHC with a (probably) very broad HPV antibody lacks the desired specificity. The authors should consider either eliminating this part of the study or reevaluation of HPV status based on more recent literature pertinent to the immunocompetent subjects and in conjunction with employing more specific methods.

#This was a very meaningful suggestion. I completely agree with the reviewer. References to HPV in the conclusion section have been deleted.

L244 In the Discussion, we added the following:

The association of SCC with HPV was not confirmed because the number of cases was small. In addition, our results may not be accurate because we did not use multiplex PCR, which is currently the most suitable genotyping method.26

2. The authors use immunohistochemistry staining to draw conclusions on the intracellular distribution of EGFR and the translocation of EGFR from the membrane into the cytoplasm. In table 2 they show that all cases of invasive SCC and 14/15 cases of in situ carcinoma show various degrees (weak in the majority of cases) of cytoplasmic staining in combination with membranous staining (14/14 in invasive SCC, 14/15 in situ carcinoma). There is no correlation between the degree/intensity of membranous vs cytoplasmic staining. Assuming that cytoplasmic staining marks translocation of EGFR from the membrane into cytoplasm is not scientifically sound.

#Thank you for your valuable suggestions. For this phenomenon, we further compared our cases with cases of benign conjunctival mass lesions. Although the number was small (n = 7), the following results were obtained.

Staining of the cytoplasm was significantly different between the benign disease group and malignant cases, which was a new finding for all conjunctival tumors. This information has been added to the Results and Discussion.

L158- seven benign conjunctival lesions ( three pinguecula, three pterygium, one dermoid cyst) were partially weak positive in conjunctival squamous epithelial cells , especially cell surface membranes(Figure 2).In addition, cytoplasmic staining are only one case. Benign cases showed scores of 1.28 for membrane staining and 0.14 for cytoplasmic staining, compared with conjunctival Scc cases, Cytoplasmic staining was statistically significant changes ( P<0.01) (Table.2).

L200- Although the number of cases examined was small, the expression of cytoplasmic staining of EGFR was significantly weak in the benign disease group. The hypothesis might be that the EGFR transition from membrane surface into cytoplasm will progress as it becomes more malignancy changes.

3. The authors claim that this is the first report to show that mutations in EGFR are associated with prognosis and treatment. They allude to the EGFR mutations in lung cancer and they show in table 3 the results of the 2 investigated mutations but there is no correlation of the 2 mutations with prognosis and treatment in ocular neoplasia.

#Thank you very much for pointing this out.

The Cox regression model was used to examine the relationship between EGFR mutations and long-term prognosis including orbital exenteration and PFS. EGFR mutations were not significantly correlated with final orbital exenteration or PFS (Tables 4 and 5). The small number of cases may be the reason why no significant difference was found. Examination of these cases suggested that mutation-positive cases also tended to have slightly advanced cancer, but no statistically significant difference was found. The following was added to the conclusion.

L264 Our study found that EGFR mutations were also found in the conjunctival Scc, but we could not obtain the results that would correlate with the final prognosis. It seems likely that further studies including further multi-institutions and an increase in the number of cases will be expected in the future.

4. The specificity of the 2 antibodies used for the 2 EGFR mutations is not addressed and no correlation with molecular studies identifying the same 2 mutations is done. However, the authors conclude that IHC can be used to identify EGFR mutations.

#This is a reasonable question.

Certainly, double testing of formalin-fixed paraffin-embedded tissue and plasma with real-time PCR for detection of EGFR mutations is more common than IHC in the actual clinical setting. According to the literature, both sensitivity and specificity seemed to be satisfactory for these two types of mutations. Reference No. 30 was added. We also described this as a limitation.

30. Kane S, Wu J, Benedettini E, Li D, Reeves C, Innocenti G, Wetzel R, Crosby K, Becker A, Ferrante M, Cheung WC, Hong X, Chirieac LR, Sholl LM, Haack H, Smith BL, Polakiewicz RD, Tan Y, Gu TL, Loda M, Zhou X, Comb MJ.. Mutation-specific antibodies for the detection of EGFR mutations in non -small-cell lung cancer. Clin Cancer Res 2009;15:3023-3028.

Reviewer #2:

Thank you for your valuable comments.

Our responses to your comments are below.

 1. Authors have first found out that EGFR have focal and diffuse staining in patients. This is really an interesting observation however, authors later never looked for its correlation with localization or mutation of EGFR in patients.

#This is a very meaningful suggestion.

We also realized that this point was a problem, and thus, we reviewed the scoring for each of the two groups. Then, we performed statistical analysis with linear regression calculation. Unfortunately, the cytoplasmic group did not show a statistically significant difference, but the diffuse EGFR group tended to have a higher cytoplasmic score (P = 0.077). The membrane group did not show a significant difference or tendency for a higher score (P = 0.214). We added this to the Results and also mentioned it in the Discussion.

L164

The correlation between EGFR stating (focal or diffuse) and EGFR localization (cytoplasmic staining group ) did not show a statistically significant difference, but the diffuse EGFR group tended to have a higher score(p=0.077). The correlation between EGFR stating (focal or diffuse) and EGFR localization (membrane group) was no significant difference and no tendency (p=0.214).

L200

Although the number of cases examined was small, the expression of cytoplasmic staining of EGFR was weak but significantly different from membrane staining. Our hypothesis is that as EGFR transitions from the membrane into the cytoplasm, malignant changes progress. In addition, a correlation between EGFR staining (focal or diffuse) and EGFR cytoplasmic staining was seen, and a higher score tended to be present in the diffuse EGFR staining group.

Although the difference was not significant in this study, intracellular transfer of EGFR in the group with diffuse staining may indicate progression, and significant findings may emerge by increasing the number of cases in the future.

L173 The relationship between EGFR mutation and EGFR stating (focal or diffuse) was determined using univariate linear regression analysis with making corrections based on age(p=0.559)

2. In figure 1A&B, authors showed representative images of local and diffuse EGFR staining. It appears that images of EGFR staining in both figures have different magnification. Similarly figure 1E captured at different magnification compared to figure 1C&D. Figure 2 have similar issue where image D have different magnification compared to rest of the images. Authors need to make sure representative images were captured at same magnification for better comparison. Authors also need add scale bar on all the microscopy images.

3. In figure 1A hematoxylin-eosin-stained section in inset looks little different from EGFR stained image. Authors also need to make sure that the image in H&E inset properly corresponds to EGFR stained image.

#Thank you for your valuable suggestions.

We added scale bars and corrected the figures as you suggested (Figure 1 and Figure 2). In one panel, the HE-stained section does not match, but the same region is photographed and shown. We are sorry for the incompleteness. 

4. In present manuscript authors also evaluated EGFR mutation using immunohistochemistry. They observed EGFR E746-A750 del and EGFR L858R expression in patients. Authors need to include this parameter for univariate correlation analysis with PFS and orbital exenteration. Also, it will be interesting to see the effect of these mutation on the localization of EGFR in tumor cell.

#Thank you for your valuable suggestions.

L180 The Cox regression model was used to examine and analyze the relationship between long-term prognosis including orbital exenteration and PFS and the clinicopathological status, EGFR staining pattern, and EGFR mutation. 

L187 The EGFR mutation was not significantly correlated with final orbital exenteration or PFS (Tables 4 and 5). The small number of cases may be the reason why no significant difference was found.

5. Authors also need to perform multivariate analysis for clinical correlation for both membrane and cytoplasmic EGFR localization in patients.

#Thank you very much for pointing this out.

We also consulted with statisticians at our university, and discussed the number of cases. Because the number of parameters was small and difficult to evaluate with multivariate analysis, we decided not to perform further analysis at this time. We appreciate your understanding.

6. References in the manuscript have different fonts and wrongly numbered. Authors need to use endnote or reference manager to add the references.

#Thank you very much for pointing this out.

We will correct the incorrect parts. Although we have requested funding for Endnote, it cannot be purchased immediately. Thank you very much.

reviewer3 

Thank you for your valuable comments.

Our responses to your comments are below.

1. The result section should be written in more detail for easier reading.

#Thank you for your valuable suggestions.

We added the text below to the Results section.

L156-

No significant difference was found between carcinoma in situ (Tis) and invasive carcinoma (Tadv) (Table 2). No significant difference was found in the score depending on the stage. On the other hand, seven benign conjunctival lesions (three pinguecula, three pterygium, one dermoid cyst) showed partial weak positive staining in conjunctival squamous epithelial cells, especially on the cell membrane (Figure 2A). EGFR expression in colon cancer for positive control (Figure.2B).　In addition, cytoplasmic staining was seen in only one case. Benign cases showed scores of 1.28 for membrane staining and 0.14 for cytoplasmic staining. Cytoplasmic staining patterns were significantly different in benign compared to SCC cases (P < 0.01) (Table 2). The correlation between EGFR staining (focal or diffuse) and EGFR localization (cytoplasmic staining group) was not significantly different, but the diffuse EGFR group tended to have a higher score (P = 0.077). No correlation was found between EGFR staining (focal or diffuse) and EGFR localization (membrane group), and no apparent tendency was observed (P = 0.214).

L170

The mutation at exon 19, EGFR E7446-A750 del, was confirmed in 8/29 (27.5%) cases, and that at exon 21, EGFR L858R point mutation, was confirmed in 2/29 (6.8%) cases with IHC (Table 3). The relationship between EGFR mutation and EGFR staining (focal or diffuse) was determined using univariate linear regression analysis with correction for age (P = 0.559).

L180

COX regression model was used to examine and analysis among clinicopathological status, EGFR staining pattern, and EGFR mutation for long-term prognosis.

L187

In addition, the EGFR mutation was not significantly correlation with final orbital exenteration and PFS(Table 5).

2. Line 154: “Analysis of EGFR intracellular staining patterns showed scores of 1.72 for membrane staining and 1.48 for cytoplasmic staining (Figure 1)”. This result is found in Table 2 not in figure 1.

#Thank you for pointing this out.

L157 We corrected the text from Figure 1 to Table 2.

3. Table 2: Revise statistics. My calculation doesn’t match the average shown in the table.

# We are sorry for this mistake.

P21 Table 2. We recalculated and now show the correct value in Table 2.

4. Is there any difference in EGFR expression localization (membrane or cytoplasm) between focal and diffuse staining?

#This is a very meaningful suggestion.

We also realized that this point was a problem, and again reviewed the scoring for each of the two groups. Then, we performed statistical analysis with linear regression calculation. Unfortunately, the cytoplasmic group did not show a statistically significant difference, but the diffuse EGFR group tended to have a higher score (P = 0.077). The membrane group did not show a significant difference or tendency (P = 0.214). We added this to the Results and also mentioned it in the Discussion.

L159-169

The correlation between EGFR stating (focal or diffuse) and EGFR localization (cytoplasmic staining group ) did not show a statistically significant difference, but the diffuse EGFR group tended to have a higher score(p=0.077). The correlation between EGFR stating (focal or diffuse) and EGFR localization (membrane group) was no significant difference and no tendency (p=0.214).

L201 Although the number of cases examined was small, the expression of cytoplasmic staining of EGFR was weak but significantly different from membrane staining. Our hypothesis is that as EGFR transitions from the membrane into the cytoplasm, malignant changes progress. In addition, a correlation between EGFR staining (focal or diffuse) and EGFR cytoplasmic staining was seen, and a higher score tended to be present in the diffuse EGFR staining group.

Although the difference was not significant in this study, intracellular transfer of EGFR in the group with diffuse staining may indicate progression, and significant findings may emerge by increasing the number of cases in the future.

5. Line 156: “Seven benign conjunctival lesions were partially weak positive in conjunctival squamous epithelial cells near basement membrane”. Specify the total number of benign samples evaluated (7 out of 7?). It would be interesting to show the score analysis of benign samples to compare to tumor samples.

#This was a very meaningful suggestion. Thank you very much. 

The staining of the cytoplasm was significantly different between the benign disease group and malignant cases, which was a new finding for conjunctival tumors. This information was added to the Results and Discussion.

L159- seven benign conjunctival lesions ( three pinguecula, three pterygium, one dermoid cyst) were partially weak positive in conjunctival squamous epithelial cells , especially cell surface membranes(Figure 2).In addition, cytoplasmic staining are only one case. Benign cases showed scores of 1.28 for membrane staining and 0.14 for cytoplasmic staining, compared with conjunctival Scc cases, Cytoplasmic staining was statistically significant changes ( P<0.01) (Table.2).

L201- Although the number of cases examined was small, the expression of cytoplasmic staining of EGFR was significantly weak in the benign disease group. The hypothesis might be that the EGFR transition from membrane surface into cytoplasm will progress as it becomes more malignancy changes.

6. Line 157: “We found no significant changes between Tis (in situ) and Tadv (SCC advanced cases) (Table 2)”. Specify what changes.

We deleted the original sentence and added the following sentence.

L156 No statistical difference was found between Tis and Tadv in the intensity of immunostaining in staining patterns.

7. Line 159: Include more information regarding EGFR E746-A750 del and EGFR L858R expression like diffuse/focal staining, membrane/cytoplasmic score. Is there any correlation between EGFR mutation and clinical data (orbital exenteration/PFS)?

#Thank you for your valuable suggestions.

The Cox regression model was used to examine and analyze the relationship between long-term prognosis including orbital exenteration and PFS and the clinicopathological status, EGFR staining pattern, and EGFR mutation. The EGFR mutation was not significantly correlated with final orbital exenteration or PFS (Tables 4 and 5). The small number of cases may be the reason why no significant difference was found.

8. Table 1: EGFR expression in tumors - diffuse staining (8) and focal staining (19). Are these numbers correct, total samples 27?

# We are sorry for this mistake.

P19 Table 1. The value was recalculated and described correctly in Table 1.

9. Figure 1 and 2: When comparing images use the same magnification. Verify if scale bars are correct.

#Thank you for your valuable suggestions.

We inserted scale bars and corrected the figures (Figure 1 and Figure 2).

---

## [Decision Letter · Decision Letter 1]

15 Jul 2020

PONE-D-20-09283R1

Expression, intracellular localization, and mutation of EGFR in conjunctival squamous cell carcinoma and the association with prognosis and treatment

PLOS ONE

Dear Dr.Tagami,

Thank you for submitting your manuscript to PLOS ONE. After careful consideration, we feel that it has merit but does not fully meet PLOS ONE’s publication criteria as it currently stands. Therefore, we invite you to submit a revised version of the manuscript that addresses the points raised during the review process.

A learned reviewer has raised a number of critical issues that need to be satisfactorily addressed by incorporating appropriate changes in the manuscript.

We look forward to receiving your revised manuscript.

Kind regards,

Sanjoy Bhattacharya

Academic Editor

PLOS ONE

Reviewers' comments:

Reviewer's Responses to Questions

**Comments to the Author**

1. If the authors have adequately addressed your comments raised in a previous round of review and you feel that this manuscript is now acceptable for publication, you may indicate that here to bypass the “Comments to the Author” section, enter your conflict of interest statement in the “Confidential to Editor” section, and submit your "Accept" recommendation.

Reviewer #1: All comments have been addressed

Reviewer #2: All comments have been addressed

Reviewer #3: (No Response)

2. Is the manuscript technically sound, and do the data support the conclusions?

Reviewer #1: Yes

Reviewer #2: Yes

Reviewer #3: Yes

3. Has the statistical analysis been performed appropriately and rigorously? 

Reviewer #1: Yes

Reviewer #2: Yes

Reviewer #3: Yes

4. Have the authors made all data underlying the findings in their manuscript fully available?

Reviewer #1: Yes

Reviewer #2: Yes

Reviewer #3: Yes

5. Is the manuscript presented in an intelligible fashion and written in standard English?

Reviewer #1: Yes

Reviewer #2: Yes

Reviewer #3: No

6. Review Comments to the Author

Reviewer #1: Comments have been addressed. The manuscript has spelling and minor grammatical errors that need to be fixed.

Reviewer #2: (No Response)

Reviewer #3: In this revised manuscript, the authors have addressed some comments raised from my previous review report. However, there are several major concerns that need to be addressed before further consideration.

1. A major revision of language and grammar is necessary for any further consideration. There are several typos, missing spaces between words, and grammatically incorrect sentences (especially in the new text added after revision) that make it hard to read. Some examples are:

Line 161 - Benign cases showed scores of 1.28 for membrane staining and 0.14 for cytoplasmic staining, compared with conjunctival Scc cases, Cytoplasmic staining was statistically significant changes

Line 166 - The correlation between EGFR stating (focal or diffuse) and EGFR localization (membrane group) was no significant difference and tendency

Line 180 - COX regression model was used to examine and analysis among clinicopathological status, EGFR staining pattern, and EGFR mutation for long-term prognosis including orbital exenteration and.PFS.

Line 188 - In addition, the EGFR 188 mutation was not significantly correlation with final orbital exenteration and PFS(Table 4 and 5).

2. Line 206 – “There was no significant difference in this study”. Specify what had no significant difference. The whole study or a specific result?

3. Line 160, the sentence “EGFR expression in colon cancer for positive control” is out of context. What is the relevance of this information to the results?

4. Line 164-167 – Regarding the correlation between EGFR staining (focal or diffuse) and EGFR localization (membrane or cytoplasm), it would be interesting to have a table showing the number of diffuse and focal staining samples in the membrane, and cytoplasm group. Between the slides with EGFR membrane localization, how many samples had focal staining and how many had diffuse staining? Add the same information for EGFR cytoplasm localization.

5. In table 3, add information (number of samples) regarding focal, diffuse, membrane, and cytoplasmic staining for each mutation.

6. In figure 3C, the EGFR L858R staining seems to be localized in the nucleus. Do you have an explanation for that?

7. PLOS authors have the option to publish the peer review history of their article (what does this mean?). If published, this will include your full peer review and any attached files.

Reviewer #1: No

Reviewer #2: No

Reviewer #3: No

---

## [Author Response · Author response to Decision Letter 1]

21 Jul 2020

Response to reviewers 

Reviewer #1: Thank you for your valuable comments. We submitted the revised version for proofreading by a native English-speaking science editor.

Reviewer #3

Thank you for your valuable comments. Our responses are below.

1. A major revision of language and grammar is necessary for any further consideration. There are several typos, missing spaces between words, and grammatically incorrect sentences (especially in the new text added after revision) that make it hard to read. Some examples are:

Line 161 - Benign cases showed scores of 1.28 for membrane staining and 0.14 for cytoplasmic staining, compared with conjunctival Scc cases, Cytoplasmic staining was statistically significant changes

Line 166 - The correlation between EGFR stating (focal or diffuse) and EGFR localization (membrane group) was no significant difference and tendency

Line 180 - COX regression model was used to examine and analysis among clinicopathological status, EGFR staining pattern, and EGFR mutation for long-term prognosis including orbital exenteration and.PFS.

Line 188 - In addition, the EGFR 188 mutation was not significantly correlation with final orbital exenteration and PFS(Table 4 and 5).

We submitted the revised version for proofreading by a native English-speaking science editor.

2. Line 206 – “There was no significant difference in this study”. Specify what had no significant difference. The whole study or a specific result?

Only the results of this statement are mentioned. We corrected the sentence as shown below.

L170“Intracellular transfer of EGFR in the group with diffuse staining may indicate progression, and although no statistical differences were observed in this study, significant findings may emerge by increasing the number of cases in the future.”

3. Line 160, the sentence “EGFR expression in colon cancer for positive control” is out of context. What is the relevance of this information to the results?

We corrected the sentence as shown below. The location was changed to line 163 to suit the context. Accordingly, we changed the location of the Figure 2 citation in the text.

4. Line 164-167 – Regarding the correlation between EGFR staining (focal or diffuse) and EGFR localization (membrane or cytoplasm), it would be interesting to have a table showing the number of diffuse and focal staining samples in the membrane, and cytoplasm group. Between the slides with EGFR membrane localization, how many samples had focal staining and how many had diffuse staining? Add the same information for EGFR cytoplasm localization.

We created a new Table 3 to address your comment.

Table3. EGFR staining and Localization 

 EGFR Focal(N=21) EGFR Diffuse（N=8) P

 Cell Membrane 1.8±0.9

 1.5±0.9 0.38

Cell cytoplasmic 1.3±0.6 1.8±0.8 0.12

p value based on the non-paired t-test.

5. In table 3, add information (number of samples) regarding focal, diffuse, membrane, and cytoplasmic staining for each mutation.

We created a new Table 4 to hold this information.

Table 4. Summary of EGFR E746-A750 del and EGFR L858R point mutations 

Mutation N = 29 (%) Age (y) Sex (M/F) T stage EGFR staining patterns (diffuse/focal) EGFR localization score (membrane/cytoplasmic)

Exon 19 EGFR E746-A750 del (N=8) 8/29 (27.5%) 75.8 3/5 T3: 4

T2: 2

Tis: 2 2/6 1.6/1.5

Exon 21 EGFR L858R point mutation (N=3) 2/29 (6.8%) 73.0 1/1 T3: 1

Tis: 1 1/2 1/3

M: Male, F: Female

6. In figure 3C, the EGFR L858R staining seems to be localized in the nucleus. Do you have an explanation for that?

Thank you for this advice. We agree that regarding the dyeability of EGFR, it is important whether or not the cell membrane is stained. It looks like it stains the nucleus, but we are unsure of the exact reason. We believe that it differs from the simple background because only the nuclei of squamous cell carcinoma are stained. 

There are the following documents. Whether or not this is meant is not convinced by this paper alone.

The receptor tyrosine kinases (RTK) of the epidermal growth factor receptor (EGFR) superfamily　in the membrane can translocate to the nucleus through different mechanisms. Nuclear RTKs regulate a variety of cellular functions, such as cell proliferation, DNA damage repair, and signal transduction, both in normal tissues and in human cancer cell. In addition, they play important roles in determining cancer response to cancer therapy.（Shao-Chun Wang et al. DOI: 10.1158/1078-0432.CCR-08-2813 2009）

---

## [Decision Letter · Decision Letter 2]

11 Aug 2020

Expression, intracellular localization, and mutation of EGFR in conjunctival squamous cell carcinoma and the association with prognosis and treatment

PONE-D-20-09283R2

Dear Dr. Tagami,

We’re pleased to inform you that your manuscript has been judged scientifically suitable for publication and will be formally accepted for publication once it meets all outstanding technical requirements.

Kind regards,

Sanjoy Bhattacharya

Academic Editor

PLOS ONE

Additional Editor Comments (optional):

Reviewers' comments:

Reviewer's Responses to Questions

**Comments to the Author**

1. If the authors have adequately addressed your comments raised in a previous round of review and you feel that this manuscript is now acceptable for publication, you may indicate that here to bypass the “Comments to the Author” section, enter your conflict of interest statement in the “Confidential to Editor” section, and submit your "Accept" recommendation.

Reviewer #2: All comments have been addressed

Reviewer #3: All comments have been addressed

2. Is the manuscript technically sound, and do the data support the conclusions?

Reviewer #2: Yes

Reviewer #3: Yes

3. Has the statistical analysis been performed appropriately and rigorously? 

Reviewer #2: Yes

Reviewer #3: Yes

4. Have the authors made all data underlying the findings in their manuscript fully available?

Reviewer #2: (No Response)

Reviewer #3: Yes

5. Is the manuscript presented in an intelligible fashion and written in standard English?

Reviewer #2: Yes

Reviewer #3: Yes

6. Review Comments to the Author

Reviewer #2: (No Response)

Reviewer #3: (No Response)

7. PLOS authors have the option to publish the peer review history of their article (what does this mean?). If published, this will include your full peer review and any attached files.

Reviewer #2: No

Reviewer #3: No

---

## [Editor Report · Acceptance letter]

13 Aug 2020

PONE-D-20-09283R2 

 Expression, intracellular localization, and mutation of EGFR in conjunctival squamous cell carcinoma and the association with prognosis and treatment 

Dear Dr. Tagami:

I'm pleased to inform you that your manuscript has been deemed suitable for publication in PLOS ONE. Congratulations! Your manuscript is now with our production department. 

Kind regards, 

on behalf of

Dr. Sanjoy Bhattacharya 

Academic Editor

PLOS ONE